# In Mild and Moderate Acute Ischemic Stroke, Increased Lipid Peroxidation and Lowered Antioxidant Defenses Are Strongly Associated with Disabilities and Final Stroke Core Volume

**DOI:** 10.3390/antiox12010188

**Published:** 2023-01-12

**Authors:** Michael Maes, Francis F. Brinholi, Ana Paula Michelin, Andressa K. Matsumoto, Laura de Oliveira Semeão, Abbas F. Almulla, Thitiporn Supasitthumrong, Chavit Tunvirachaisakul, Decio S. Barbosa

**Affiliations:** 1Department of Psychiatry, Faculty of Medicine, Chulalongkorn University, 1873 Rama 4 Rd., Phayathai Road, Pathumwan, Bangkok 10330, Thailand; 2Cognitive Fitness and Technology Research Unit, Faculty of Medicine, Chulalongkorn University, Bangkok 10330, Thailand; 3Department of Psychiatry, Medical University of Plovdiv, 4000 Plovdiv, Bulgaria; 4Research Institute, Medical University Plovdiv, 4000 Plovdiv, Bulgaria; 5Deakin University, IMPACT-the Institute for Mental and Physical Health and Clinical Translation, School of Medicine, Barwon Health, Geelong, VIC 3220, Australia; 6Health Sciences Graduate Program, Health Sciences Center, State University of Londrina, Londrina 86057-970, PR, Brazil; 7Medical Laboratory Technology Department, College of Medical Technology, The Islamic University, Najaf 54001, Iraq

**Keywords:** antioxidants, biomarkers, inflammation, neuroimmune, oxidative stress, physiological stress

## Abstract

In acute ischemic stroke (AIS), there are no data on whether oxidative stress biomarkers have effects above and beyond known risk factors and measurements of stroke volume. This study was conducted in 122 mild-moderate AIS patients and 40 controls and assessed the modified ranking scale (mRS) at baseline, and 3 and 6 months later. We measured lipid hydroperoxides (LOOH), malondialdehyde (MDA), advanced oxidation protein products, paraoxonase 1 (PON1) activities and PON1 Q192R genotypes, high density lipoprotein cholesterol (HDL), sulfhydryl (-SH) groups), and diffusion-weighted imaging (DWI) stroke volume and fluid-attenuated inversion recovery (FLAIR) signal intensity. We found that (a) AIS is characterized by lower chloromethyl acetate CMPAase PON1 activity, HDL and -SH groups and increased LOOH and neurotoxicity (a composite of LOOH, inflammatory markers and glycated hemoglobin); (b) oxidative and antioxidant biomarkers strongly and independently predict mRS scores 3 and 6 months later, DWI stroke volume and FLAIR signal intensity; and (c) the PON1 Q192R variant has multiple effects on stroke outcomes that are mediated by its effects on antioxidant defenses and lipid peroxidation. Lipid peroxidation and lowered -SH and PON1-HDL activity are drug targets to prevent AIS and consequent neurodegenerative processes and increased oxidative reperfusion mediators due to ischemia-reperfusion injury.

## 1. Introduction

One in every four people will have a stroke in their lifetime, and there are more than 100 million stroke survivors worldwide [1,2]. Stroke is the second leading cause of death and disability, accounting for approximately 13 million new cases each year, with approximately one-third of stroke victims dying and another third becoming chronically incapacitated and requiring permanent residential care [2,3,4]. Stroke has major socioeconomic effects and is the second leading cause of hospital care expenditures among people with cardiovascular illnesses [5]. According to statistical projections, the aging population will cause this number to skyrocket over the next few decades [6].

Acute ischemic stroke (AIS), which accounts for 87% of all strokes worldwide, may be defined as a rapid loss of brain function caused by a significant disruption of blood and oxygen flow in the cerebral arteries due to an embolism or thrombus. AIS results from the cumulative, long-term effects of irreversible factors, including increasing age, sex, genetic factors, and ethnicity, as well as modifiable risk factors, including increased systolic and diastolic blood pressure, atrial fibrillation, transient ischemic attack, type 2 diabetes mellitus, excess alcohol consumption, smoking, sedentary lifestyle, increased body mass index (BMI), increased levels of glucose, triglycerides, low-density lipoprotein (LDL) cholesterol, and lowered levels of high-density lipoprotein (HDL) cholesterol [7,8,9,10,11].

AIS results in cerebral ischemia and leads to necrotic and excitotoxic cell death in the ischemic core [12,13], whereby highly connected immune and hemostasis subnetworks and activation of intracellular pathways may lead to post-stroke death [11]. Following AIS, nitro-oxidative and neuroinflammatory (microglia activation) processes are critically involved in the primary insult, leading to neuroapoptosis, neurotoxicity, necrosis or autophagy and consequent brain-cell death in the stroke core and secondary neurodegenerative processes in the penumbra [11,12,13,14]. Ischemia and the consequent excitotoxicity and reperfusion are associated with free radical-mediated reactions contributing to neuronal cell death [15], whilst reperfusion is accompanied by secondary neurotoxic responses induced by a second burst in reactive oxygen species (ROS) and pro-inflammatory cytokines, such as interleukin (IL)-6, IL-1β and tumor necrosis factor (TNF)-α [11,12,13].

Immunological and oxidative mediators produced in the brain can diffuse into the systemic circulation, triggering an immune-inflammatory response in the periphery [16]. Experimental stroke models elicit a peripheral immune-inflammatory response which peaks four hours after stroke and precedes the peak in neuroinflammation twenty hours later [17]. This peripheral response to AIS is characterized by (a) elevated levels of pro-inflammatory cytokines and inflammatory biomarkers, including C-reactive protein (CRP), and white blood cell (WBC) numbers, (b) increased ROS and nitro-oxidative stress with elevated lipid peroxidation (lipid hydroperoxides or LOOH) and nitric oxide metabolite (NOx) levels, (c) decreased antioxidant defenses, including decreased paraoxonase 1 (PON1), especially chloromethyl phenylacetate CMPAase activity, and 25-OH vitamin D (25(OH)D [11,13,17,18,19,20,21,22,23]. Moreover, activation of immune-inflammatory and oxidative pathways in peripheral blood plays a crucial role in neurological outcomes of AIS [17], as altered endothelial cell functions and increased ROS may disrupt the blood brain barrier (BBB), contributing to the infiltration of blood-derived M1 macrophages, activated T cells and neutrophils, thereby aggravating neuroinflammation and neurotoxicity [12,13,16]. The effects of the PON1 Q192R gene on the outcome of a stroke are partially mediated by their effect on two distinct catalytic sites, specifically CMPAase vs arylesterase (AREase) activities [21]. However, it is unknown if the PON1 Q192R gene may influence the outcome of a stroke by influencing antioxidant defenses, lipid peroxidation, neurotoxicity and stroke volume.

Recent studies show that the cumulative effects of peripheral immune-inflammatory, oxidative and nitrosative, and metabolic biomarkers are associated with AIS with adequate predictive power. For example, using the cumulative effects of LOOH, NOx, fasting blood glucose (FBG), WBC count, IL-6, and 25(OH)D, and classical risk factors such as male sex and systolic blood pressure, 89.4% of AIS patients were correctly classified with a sensitivity of 86.2% and specificity of 93.0% [19]. Moreover, elevated scores of baseline functional disability as assessed with the modified Rankin score (mRS, threshold ≥ 3) were associated with increased WBC, high sensitivity CRP (hsCRP), FBG, IL-6, and ferritin, whilst 25.0% of the variance in mRS values 6 months later was predicted by the regression on HDL-cholesterol and 25(OH)D (both inversely) and FBG, age and sedimentation rate (all positively). Moreover, 34.7% of the variance in baseline score of the National Institutes of Health Stroke Scale (NIHSS) score was explained by PON1 status and HDL-cholesterol and hypertension [21].

There is also evidence that AIS with larger infarct sizes, as measured by diffusion-weighted imaging (DWI), has a poorer clinical outcome following intravenous thrombolysis, whilst the quantification of infarction evolution may permit the determination of treatment efficacy [24,25,26,27,28,29,30]. Within minutes following artery closure, there is an increase in DWI signal which reveals the infarcted core in AIS patients demonstrating the true minimum extension of infarction with high interobserver agreement, sensitivity, and specificity [31]. There is a steady increase in fluid-attenuated inversion recovery (FLAIR) signal intensity in the infarcted region, which can be assessed several hours after the AIS and is associated with the appearance of vasogenic oedema and predicts a poor outcome [32,33,34,35]. In addition, FLAIR also detects white matter hyperintensities (WMHs) as the result of atherosclerosis processes or small vessel disease [36].

However, there are no data on whether lipid and protein oxidation and antioxidant biomarkers including the PON1 Q192R gene are associated with short- and medium-term outcomes of AIS above and beyond the effects of traditional risk factors, immune and metabolic biomarkers, and DWI and/or FLAIR MRI assessments; or whether oxidative and antioxidant biomarkers are associated with DWI stroke volume or FLAIR lesions. Hence, the present study was conducted to delineate whether AIS and disabilities (as assessed with NIHSS and mRS) at baseline, and 3 and 6 months later, and DWI stroke volume and/or FLAIR signal intensity, are associated with lipid and protein oxidation (LOOH, MDA and AOPP assays) and antioxidant (PON1 activities, PON1 Q192R genotypes, HDL and sulfhydryl or -SH groups) biomarkers and whether these markers have an impact above and beyond classical risk factors (e.g., age, body mass index or BMI, hypertension, cardiovascular disease), immune biomarkers (including hsCRP, WBC counts and neutrophil/lymphocyte ratio (NLR)), metabolic variables (FBG and glycated hemoglobin (HbA1c), and athergenicity indices.

## 2. Materials and Methods

### 2.1. Participants

This study recruited 122 patients with AIS and 40 healthy volunteers as participants. Between October 2019 and September 2020, the Stroke Unit at King Chulalongkorn Memorial Hospital admitted patients. Controls were recruited from the same catchment area by word of mouth (Bangkok province, Thailand). Patients and controls were subjected to basic blood sampling and clinical assessments within eight hours of admission, and some patients were checked again three and six months later. Eligible patients had AIS as indicated by localized neurological signs and symptoms of vascular origin that persisted for more than 24 h, as established by clinical examination by an experienced neurologist, and brain CT scan measures indicative of AIS. The exclusion criteria for controls were any axis-1 mental condition and a positive family history of depression and mood disorders in first-degree relatives. The exclusion criteria for patients and controls were as follows: (a) neurological disorders, such as Alzheimer’s and Parkinson’s disease, and multiple sclerosis; (b) neuropsychiatric disorders, such as schizophrenia, major depression, bipolar disorder, obsessive-compulsive disorders, post-traumatic stress disorder, psycho-organic syndrome, and substance use disorders; (c) autoimmune and immune disorders, such as systemic lupus erythematosus, rheumatoid arthritis, inflammatory bowel disease, psoriasis, type 1 diabetes mellitus, COPD; (d) renal or liver failure, cancer, and infectious disease including HIV and hepatitis B infection; and (e) use of antidepressants or other psychoactive drugs. The patient exclusion criteria were as follows: (a) patients who were unable to cooperate with the verbal clinical interview because they were unconscious, aphasic, or had severe cognitive impairments; and (b) patients who experienced a aemorrhagic stroke or transient ischemic attack.

The study was authorized by the Institutional Review Board of Chulalongkorn University’s Faculty of Medicine in Bangkok, Thailand (IRB no. 62/073), which complies with the International Guideline for Human Research Protection, as required by the Declaration of Helsinki, the Belmont Report, the CIOMS Guidelines, and the International Conference on Harmonization in Good Clinical Practice. Prior to inclusion in the study, all participants and guardians of the patients submitted written informed consent.

### 2.2. Clinical Assessments

Semi-structured interviews were used to collect sociodemographic data such as gender, age, years of education, employment, and marital status, as well as clinical data including vascular risk factors such as prior stroke, hypertension, atrial fibrillation, ischemic heart disease, dyslipidaemia, and type 2 diabetes mellitus (T2DM). The TOAST criteria were utilized to determine the subtype of ischemic stroke, which comprises large artery atherosclerosis (LAAS), lacunar infarction (LAC), cardioembolic infarction (CEI), stroke of other identified aetiology (ODE), and stroke of unknown aetiology (UDE) [37]. Within 24 h of admission, the NIHSS and mRS were administered to all subjects (baseline values). The mRS score was re-evaluated after three and six months of follow-up. The NIHSS measures the severity of a stroke by evaluating important domains such as language, dysarthria, degrees of awareness, facial palsy, sensation, inattention, arm and leg motor drift, limb ataxia, visual fields, and extraocular movements [38]. A high NIHSS score implies greater impairment in physical functioning after a stroke [39], with a score of 1–4 suggesting a minor stroke and a score of 5–10 indicating a moderate stroke. After a stroke, the mRS is routinely employed in clinical practice to identify the degree of disability, global impairment, or reliance [40]. No symptoms (0 score), some symptoms but no disability (1), modest (2), moderate (3), moderately severe (4), and severe disability (5) are scored on the mRS (5). The mRS scores classify the stroke outcome as good (mRS < 3) and poor (mRS ≥ 3) [41]. In the present work, we developed a global indicator of stroke severity as the z transformation of the mRS score (z mRS) + z NIHSS score.

### 2.3. Assays

Blood for the analysis of nitro-oxidative stress, immune and metabolic indicators was drawn at 8:00 a.m. following an overnight fast and within 48 h after admission into hospital. Aliquots of serum were kept at 80 °C until thawed for analysis. The O&NS biomarkers measured include LOOH, MDA, AOPP, NOx, -SH groups, chloromethyl phenylacetate CMPAase, and arylesterase (AREase). LOOH was measured by chemiluminescence in a Glomax Luminometer (TD 20/20), in the dark, at 30 °C for 60 min [42,43], and the findings were represented in relative light units. Complexation with two molecules of thiobarbituric acid and MDA estimation through high-performance liquid chromatography (HPLC Alliance e2695, Waters’, Barueri, SP, Brazil) were used to assess MDA levels [44] (Column Eclipse XDB-C18, Agilent, Santa Clara, CA, USA; mobile phase composed of 65% potassium phosphate buffer (50 mM pH 7.0) and 35% HPLC grade methanol; flow rate of 1.0 mL/min; temperature of 30 degrees Celsius; wavelength of 532 nm). Based on a calibration curve, the MDA concentration in the samples was determined and is given as mmol of MDA per mg of protein. At 340 nm, AOPP was quantified in mM of equivalent chloramine T using a microplate reader (EnSpire, Perkin Elmer, Waltham, MA, USA) [45,46]. NO metabolites (NOx) were determined by measuring nitrite and nitrate concentrations in a microplate reader (EnSpire^®^, Perkin Elmer, USA) at 540 nm [47], and the results are reported as M. The -SH groups were evaluated using a microplate reader (EnSpire^®^, Perkin Elmer, USA) at 412 nm, and the results were expressed in millimolar (M) [48,49]. All inter-assay coefficients of variation were <10.0%). PON1 status was determined by AREase and CMPAase activity and PON1 Q192R genotypes in the present investigation [21,50,51,52,53]. We investigated the rate of phenylacetate hydrolysis at low salt concentration by measuring the activity of AREase and CMPAase (Sigma, St. Louis, MO, USA)-ase [21,50,51,52,53]. A Perkin Elmer^®^ EnSpire model microplate reader (Waltham, MA, USA) was used to monitor the rate of phenylacetate hydrolysis at a constant temperature of 25° for 4 min (16 measurements with 15 s between each reading). The activity was expressed in units per millilitre (U/mL) based on the phenyl acetate molar extinction coefficient of 1.31 mMol/L cm^−1^. CMPA and phenyl acetate were also utilized to stratify the functional genotypes of the PON1Q192R polymorphism (PON1 192Q/Q, PON1 192Q/R, and PON1 192R/R) (Sigma, PA, USA). The phenylacetate reaction is conducted with high salt concentrations, which partially inhibits the R allozyme’s activity, allowing for a sharper distinction between the three functional genotypes. The O&NS biomarkers were assessed at baseline and 3 months later.

hsCRP, white blood cell (WBC) count, and neutrophil/lymphocyte ratio (NLR) are the immunological biomarkers that were examined at baseline. Using Alinity C (Abbott Laboratories, Chicago, IL, USA; Otawara-Shi, Tochigi-Ken, Japan) with an immunoturbidimetric approach to quantify hsCRP and a flow cytometric method (semiconductor laser) to assess WBC count, these biomarkers were evaluated. Our laboratory determined the following inter-assay coefficients of variation: hsCRP: 2.03%, WBC count: 2.0%, neutrophil%: 1.9%, and lymphocyte%: 3.5%. FBG, HbA1c, total and HDL-cholesterol, and triglycerides are the metabolic indicators that were assessed at baseline. The inter-assay coefficients of variation for FBG and HbA1c utilizing the Alinity C (Abbott Laboratories, USA; Otawara-Shi, Tochigi-Ken, Japan) in an enzymatic technique were 1.7% and 1.5%, respectively. Using the Alinity C (Abbott Laboratories, USA; Otawara-Shi, Tochigi-Ken, Japan) with enzymatic (total cholesterol), accelerator selective detergent (HDL cholesterol), and glycerol phosphate oxidase (triglyceride) techniques, total and HDL cholesterol and triglycerides were measured. The coefficients of variation for total cholesterol, HDL cholesterol, and triglycerides were 2.3%, 2.6%, and 2.3%, respectively.

DWI sequence was acquired with TR 4000 ms, TE 90 ms, 25–28 slices, slice thickness 5.0 mm, matrix 128 × 128 and FOV 230 × 230. Regions with abnormal signal were manually traced in each slice and areas (in cubic millimetres) were calculated in Dicom viewer software. The brain images were segmented into four areas using the longitudinal fissure and the frontal/dorsal sulcus. The areas were summed up and multiplied by slice thickness. A neuropsychiatrist specializing in brain imaging measured the FLAIR pictures (divided into four sections as described above), which were subsequently validated by a neuroradiologist (WMH volume is expressed in cubic millimetres). The contrast value of each patient’s FLAIR image was adjusted to increase the visibility of WMHs and other tissue. Within the brain tissue border, the application computed the area of WMHs relative to the background and an ellipse-shaped boundary was used to calculate the brain area without the presence of the skull. By multiplying the WMH area by the slice thickness of the scanned images, the volume of WMHs was determined. The semi-automated procedure lowers any bias introduced by the manual process. The ratio of WMHs to total brain volume was used to correct the WMH data for total cerebral volume. The T1-weighted images were submitted to MRIcron software [54] and brain extraction was utilized to remove skull and extra-cerebral soft tissues in order to measure the total volume of the brain. A 3D overlay volume of interest was produced with the intensity adjusted to encompass both grey and white matter, and the overlay volume was computed in centimetres cubed. We utilized Philips Ingenia Elition 3.0 Tesla MRI equipment from Philips Healthcare in Best, Netherlands. Image J, an MRI analysis calculator, was utilized to process the imaging data [55]. MRI scanning was performed 1–2 weeks after hospital admission.

Based on the assays, we computed different z unit-weighted composite score indices of: (a) antioxidant capacity of the CMPAase-HDL complex—z transformation of CMPAase (zCMPAase) + zHDL); (b) the disequilibrium between CMPAase and AREase activity—zCMPAase − zAREase; (c) immune activation: zhsCRP + zWBC + zNLR (zIMMUNE); c) neurotoxicity (zNT) − zLOOH + zhsCRP + zNLR + zHbA1c; (d) antioxidant (zANTIOX) activity—zSH groups + zCMPAase + zHDL cholesterol; and (e) NT/ANTIOX ratio—zNT − zANTIOX. Total brain lesions were computed as zFLAIR + zDWI lesion volumes.

### 2.4. Statistics

Pearson’s product-moment correlation coefficients were employed to check for correlations between two scale variables. We utilized analysis of variance to examine the differences in continuous variables between groups (ANOVA). When applicable, the Chi-square test or Fisher exact (Fisher-Freeman-Halton Exact) test were applied to check associations between categories. To normalize the distribution of variables and account for the heterogeneity in variance across research groups (as assessed by the Levene test), continuous data were converted using logarithmic (Log) or square root transformations (e.g., LOOH, hsCRP, NLR, CMPAase in log transformation). The statistical method used to examine repeated measurements (from baseline to months 3 and 6) was generalized estimating equations (GEE), with the NIHSS or mRS values from baseline, months 3 or 6 as the dependent variable and the changes in biomarkers from baseline to month 3 as explanatory variables, while allowing for the effects of age, sex, BMI, and medical comorbidities. Using multiple regression analysis, including the forward stepwise approach, the most relevant biomarkers for predicting rating-scale scores while accounting for the impact of demographic variables were identified. R^2^ variance (also used as effect size), homoscedasticity (as determined by the White and modified Breusch-Pagan tests), collinearity and multicollinearity (as assessed with the variance inflation factor > 4 or tolerance < 0.25, and the condition index and variance proportions in the collinearity diagnostics table), and multivariate normality (Cook’s distance and leverage), were all examined. To overcome collinearity issues or reduce the feature numbers, we aggregated predictors when necessary, for instance by creating z-unit-based composite scores as described above. In addition, we utilized an automated approach with a *p*-to-entry of 0.05 and a *p*-to-remove of 0.06. Automatic binary logistic regression analysis (*p*-to-enter = 0.05) was employed to delineate the significant predictors of a binary variable (e.g., hypertension, yes or no) and the Nagelkerke pseudo-R^2^ was used as an effect size. All regression analysis results were bootstrapped with 5000 samples, and the results are displayed if they do not concur. All statistical tests were two-tailed and used a significance level of *p* < 0.05. All statistical tests were performed using version 28 of IBM SPSS Windows 10.

We investigated the prediction of AIS versus controls (output variables) and IMO&NS data, traditional risk factors, BMI, age, and sex using multilayer perceptron neural network models. We developed an automated feedforward architecture model with one to two hidden layers using a maximum of eight nodes, maximal 250 epochs, and minibatch training. The halting condition consisted of one consecutive step with no additional error term reduction. We determined the error, relative error, percentage of incorrect classifications, area under the receiver operating characteristic curve (AUC ROC), and (relative) relevance of the input variables. The sensitivity and specificity of the model, as well as its accuracy, were recorded in the confusion matrix table. Towards this end, participants were first divided into three groups: (a) a training sample to estimate model parameters (46.67 percent of all cases); (b) a testing sample to prevent overtraining (20 percent of all cases); and (c) a holdout set to evaluate predictive validity (33.33 percent of all patients). Neural network was performed using version 28 of IBM SPSS Windows 10.

The causal links between the biomarkers, hypertension, and the phenome of stroke were investigated using partial least squares (PLS) path analysis (SmartPLS) [56,57]. PLS path analysis was only performed if the inner and outer models met the following specified quality standards: (a) the output latent vectors demonstrate accurate construct and convergence validity as indicated by average variance explained (AVE) > 0.5, Cronbach’s alpha > 0.7, composite reliability > 0.8, and rho A > 0.8, (b) all outer loadings are >0.6 at *p* < 0.001, (c) confirmatory tetrad analysis (CTA) showed that the model is not miss-specified as a reflective model, (d) the model’s prediction performance is adequate using PLSPredict, and (e) the overall model fit, namely the standardized root square residuals (SRMR) is accurate, namely < 0.08. If all of the above-mentioned model quality criteria were satisfied, we conducted complete PLS path analysis with 5000 bootstrap samples, produce the path coefficients (with exact *p*-values), and additionally computed the specific and total indirect (that is, mediated) effects as well as the total effects.

## 3. Results

### 3.1. Demographic and Clinical Data

Table 1 shows that there are significant differences in demographic and clinical data (age, sex, education, BMI, TUD) between controls and patients divided into those with mild and moderate AIS. The latter distinction was made using the q75 cut-off value of the z-unit-based composite score z mRS + z NIHSS. Thus, the moderate group (*n* = 37) shows increased mRS and NIHSS scores as compared with the mild AIS group. There were no significant differences in TOAST phenotypes and comorbidities between the mild and moderate AIS groups. Patients allocated to the moderate group showed higher hsCRP, NLR, total FLAIR and DWI lesions as compared with controls and mild AIS. The WBC count, z IMMUNE score, total FLAIR/volume, total MRI lesion volume, and HDL-cholesterol were significantly different between the three groups, with the latter decreasing from controls to mild AIS and moderate AIS, whilst the other markers increased. Both atherogenic indices were significantly higher in patients than in controls.

### 3.2. Associations between OS/ANTIOX Status and the Diagnostic Groups

Table 2 shows the associations between AIS groups and the NOS/ANTIOX biomarkers (results of univariate GLM analysis). -SH groups, CMPAase, zCMPAase-zAREase, zCMPAse + zHDL, NOx and ANTIOX were significantly lower in both AIS groups as compared with controls (both baseline and at3 months). In baseline conditions, AREase was higher in AIS, whereas after 3 months AREase was significantly lower in AIS than in controls. LOOH, NT and the NT/ANTIOX ratio were significantly different between the three groups and increased from controls to mild AIS to moderate AIS. 

Table 3 and Figure 1 show the results of a first neural network analysis with AIS versus controls as input variables. We trained the feedforward network with two hidden layers and used a sum of squares error term, which was lowered by training (from 2.990 to 0.843). The percentage of incorrect classifications is quite consistent among the training, testing (4.5%) and holdout (6.5%) sets, indicating that the model is very replicable and not overtrained. The AUC ROC curve was 0.991, with sensitivity of 95% and specificity of 100% in the holdout set. Figure 1 shows that LOOH, zCMPAase-zAREase, WBC count, and -SH groups had the greatest predictive power of the model, followed at a distance by AREase, NLR and zCMPAase + zHDL, and again at a distance by HbA1c, hsCRP and CMPAase. Figure 2 shows a second model in which we entered the relevant composites (NT, ANTIOX, NT/ANTIOX ratio), with classical risk factor for AIS. Table 3 shows the neural network characteristics. The top-six most important variables with the largest predictive power of the model were zNT-zANTIOX, hypertension, and zNT, followed at a distance by the Castelli risk index 1, prior stroke, dyslipidemia, BMI, and ANTIOX. Other variables were less relevant and thus omitted from the final model (e.g., age, sex). The AUC ROC curve was 0.984, with sensitivity of 97% and specificity of 92.3% in the holdout set.

Using logistic regression, we examined the associations between the biomarkers and hypertension and dyslipidemia. Hypertension was best predicted by zCMPAAse-zAREase 9 (W = 8.49, df = 1, *p* = 0.004) and Castelli 1 index (W = 9.83, df = 1, *p* = 0.002) (χ^2^ = 25.26, df = 2, *p* < 0.001, Nagelkerke = 0.191). Dyslipidemia was associated with zCMPAase-zAREase (W = 8.74, df = 1, *p* = 0.003, χ^2^ = 8.74, df = 2, *p* = 0.003, Nagelkerke = 0.080).

### 3.3. Effects of Time on the OS/ANTIOX Data

The effects of time (from baseline to 3 months later) on OS/ANTIOX biomarkers were examined in the patients. We used GEE, repeated measures, with the biomarkers as repeated measures and with the effects of time and the time × group (mild versus moderate) interaction as explanatory variables (for the PON1 activities we also entered the PON1 genotype). Table 2 shows the values of baseline conditions and three months later in both mild and moderate AIS. There were no significant time and time × group effects on -SH groups, AOPP and MDA. CMPAase (W = 16.81, *p* < 0.001), AREase (W = 100.12, *p* < 0.001) and LOOH (W = 6.27, *p* = 0.012) were significantly lower at 3 months than at baseline conditions, whilst NOx levels were significantly higher at month 3 than at baseline conditions. There were no significant time × group interactions for any of the biomarkers.

### 3.4. Intercorrelations

Table 4 shows the intercorrelation matrix between the NIHSS and mRS data (the latter also 3 months later) and the baseline OS/ANTIOX biomarkers. Both baseline NIHSS and mRS scores were significantly and inversely correlated with -SH groups, CMPAase, zCMPAase-zAREase, HDL, zCMPAase + zHDL, and ANTIOX, and positively with AREase, LOOH, zNT, zNT-zANTIOX and zIMMUNE. Baseline -SH groups, CMPAase, zCMPAase-zAREase, HDL, zCMPAase + zHDL, and ANTIOX, were significantly and inversely correlated with mRS scores at month 3, whilst LOOH, zNT, zNT-zANTIOX and zIMMUNE were positively correlated. FLAIR lesions were inversely associated with CMPAase, zCMPAase-zAREase, HDL, zCMPAase + zHDL, and zANTIOZ, and positively with zNT, zNT-zANTIOX and zIMMUNE. DWI stroke volume was significantly correlated with -SH groups, CMPAase, zCMPAase-zAREase, cMPASe + zHDL, AOPP and zANTIOX and positively with zNT, zNT-zANTIOX and zIMMUNE.

### 3.5. Results of Multiple Regression Analyses

Table 5 model #1 shows that 32.3% of the variance in the baseline NIHSS score was explained by the regression on HDL, -SH groups, and CMPAase (inversely) and AREase, LOOH, and NLR (positively). Classical risk factors were not significant in this regression, including hypertension, previous stroke, dyslipidemia, age, sex, BMI, smoking, Castelli and AIP indices. We found that 53.5% of the variance in basal mRS (model #2) was explained by HDL, -SH groups, and CMPAase (inversely) and AREase, LOOH, WBC counts and NLR (positively), whilst hypertension showed a trend towards a significant effect (*p* = 0.064).

We re-ran the analyses using the different composite scores and found that 25.6% of the variance in the baseline NIHSS score (model #3) was explained by zNT-zANTIOX and zCMPAase-zAREase. Again, none of the classical markers was significant. Figure 3 shows the partial regression of baseline NIHSS score on zNT (after adjusting for age, sex, BMI, and smoking). Model #4 shows that 52.9% of the variance in baseline mRS values was explained by zNT-zANTIOX, zIMMUNE and hypertension (all positive), and zCMPAase-zAREase (inversely). Figure 4 shows the partial regression of baseline NIHSS score on zNT (after adjusting for age, sex, BMI, and smoking). We detected that 43.0% of the variance in mRS at 3 months was explained by NT, zCMPAase-zAREase and previous stroke (model #5). Figure 5 shows the partial regression of mRS at 3 months on zNT (after adjusting for age, sex, BMI, and smoking). 24.9% of the variance in mRS at 6 months was explained by dyslipidemia and zCMPAase-zAREase and an increase in AOPP from baseline to three months later. 

### 3.6. O&NS and MRI Measurements

Table 6 examines whether O&NS biomarkers have an impact on NIHSS/mRS scores, above and beyond the effects of FLAIR/DWI measurements and classical risk factors. Model #1 shows that 46.5% of the variance in baseline NIHSS was predicted by DWI left posterior and zIMMUNE (both positively) and zCMPAase-zARease (inversely). Model 2 shows that 64.7% of the variance in baseline mRS was explained by zNT-zANTIOX, hypertension and DWI measurements (all positively). The mRS scores at 3 months were best predicted (model #3) by zNT-zANTIOX, FLAIR and DWI measurements, and dyslipidemia (all positively). 

Consequently, we have also examined the associations between DWI/FLAIR measurements and O&NS biomarkers while allowing for the effects of the classical risk factors. We found that 37.8% of the variance in total DWI volume was explained by zNT-zANTIOX (positively) and zCMPAase-zAREase (inversely) and that 47.4% in the FLAIR lesions volume was explained by previous stroke and zNT-zANTIOX (both positively) and zCMPAase-zAREase (inversely).

### 3.7. Results of PLS Analysis

Figure 6 shows the final PLS model which entered AIS as a reflective model (latent vector extracted from mRS and NIHSS and the ordinal groups shown in Table 1) and classical risk factors (RISK, entered as a composite score) as output variables and all input variables as single indicators. With a SRMR of 0.028, the overall model quality was more than adequate. The outer AIS model showed appropriate convergence and construct-reliability-validity values for the AIS (AVE = 0.914, Cronbach alpha = 0.953, rho A = 0.973, and composite reliability = 0.969), whereas all outer loadings were greater than 0.921 at *p* < 0.0001. The RISK composite showed the highest weight for hypertension and prior stroke, whereas dyslipidaemia was not really significant. PLSPredict showed that the AIS construct indicator Q2 predict values of the manifest and latent vector variables were positive, indicating that the prediction error was smaller than the most conservative benchmark. CTA demonstrated that the AIS latent vector was not improperly employed as a reflective model. Figure 6 shows that 45.3% of the variance in the AIS factor was explained by zNT, zANTIOX, AREase, hypertension, WBC count, and the RISK composite. There were significant effects of the PON1 Q192R genotype (additive model) on CMPASe, -SH groups and AREase (indicating that the RR genotype increases CMPAase but decreases -SH groups and AREase). All specific indirect effects were significant in this path analysis. The PON1 QQ genotype had a significant positive total effect on LOOH (t = 2.02, *p* = 0.022), zNT (t = 2.01, *p* = 0.022), and AIS (t = 3.38, *p* < 0.001), whereas the RR genotype has a significant negative total effect on LOOH (t = −2.03; *p* = 0.021), NT (t = −2.00; *p* = 0.023) and AIS (t = −3.43, *p* < 0.001). We also entered a latent vector based on the FLAIR, DWI and zFLAIR + zDWI in the same analysis and found that this latent vector (reflecting total brain lesions) together with zNT, zANTIOX, and WBC count explained 50.2% of the variance in the AIS latent vector. Moreover, the effects of the RISK factor were completely mediated by the lesion latent vector (t = 3.23, *p* = 0.001). This model showed significant total effects of the Q192R genotype (additive model with QQ = 0, QR = 1 and RR = 2) on zANTIOX (t = +2.35, *p* = 0.009), LOOH (t = −2.08, *p* = 0.019), neurotoxicity (t = −2.06, *p* = 0.020), brain lesions (t = −2.04, *p* = 0.04) and AIS outcome (t = −3.92, *p* < 0.001). The QQ genotype had a significant total effect on brain lesions (t = +2.03, *p* = 0.021) and AIS outcome (t = 1.97, *p* = 0.024).

## 4. Discussion

### 4.1. Oxidative Stress in AIS

The major findings of this study are that (a) AIS is characterized by increased oxidative stress, lowered antioxidant defenses, and increased neurotoxicity and that those biomarkers have more impact than immune-inflammatory and metabolic biomarkers and exert effects above and beyond classical risk factors and MRI stroke volume measurements; (b) the same oxidative/antioxidant biomarkers are strongly associated with AIS-induced disabilities and predict the mRS scores 3 and 6 months later; and (c) the best prediction of AIS and disabilities was obtained when combining oxidative, antioxidant, inflammatory biomarkers with MRI measurements, whilst the classical risk factors, except hypertension, did not have an impact after considering the blood biomarkers.

In the present study, increased LOOH was the most important biomarker of AIS and baseline disabilities, whereas MDA and AOPP were not significantly associated, and NOx was even decreased in AIS patients. Prior research has demonstrated that AIS, including in animal models, is associated with elevated levels of LOOH [13,58]. Others have reported that MDA was significantly increased in AIS [13,58,59,60,61]. The different results between the former studies (reporting increased MDA) and our study (reporting no changes in MDA despite very high LOOH values) may be explained by a biphasic response in MDA production. Thus, Scharpe et al. [62] found no significant increases in MDA (day of admission into hospital) whilst MDA levels were significantly higher 48 h later. Nonetheless, while LOOH was greatly enhanced in AIS, we found no alterations in the more severe indicators of oxidative damage to proteins (AOPP) and lipid peroxidation (MDA production) in baseline conditions. These oxidative damage indicators arise in neuropsychiatric illness, particularly in those with severe subtypes [63,64]. Because we covered mild and moderate cases, we may have missed any anomalies in these biomarkers.

Some authors reported increased NOx production [58,65,66,67] or a biphasic response in NOx levels with increased levels within the first 72 h and a decline until 96 h [60]. Interestingly, the increased NOx (and MDA) levels, which were assayed within the first 48 h post-stroke, were significantly corelated with the Canadian Neurological Scale scores [67]. When neuronal nitric oxide synthase (nNOS) and inducible NOS (iNOS) are activated in excess, NO transforms from a physiological neuromodulator to a neurotoxic component [68,69]. The latter neurotoxic effects may induce neuronal apoptosis and enlarge the cerebral lesions in association with a poorer outcome [65,70,71]. It should be underscored, however, that the lowered NOx levels in at least two AIS studies may indicate increased consumption of NO for nitration, nitrosation and nitrosylation processes, and the production of peroxynitrite, which all may contribute to increased neurotoxicity [72]. Previously, it was shown that immune (e.g., CRP, IL-6) and metabolic biomarkers (e.g., BMI and FBG) increase stroke risk and contribute to a worse outcome (see reviews: [18,19,20]).

Nevertheless, in the present study, the effects of oxidative/antioxidant markers on AIS and disabilities were much more important than those of the inflammatory (hsCRP, NLR, WBC) and metabolic (BMI, FBG, HbA1c) biomarkers. In the current study, all effects of hsCRP disappeared after considering the impact of the oxidative biomarkers, whilst NLR affected baseline NIHSS and mRS scores with a smaller effect size and was not related to the mRS scores 3 or 6 months later. NLR is a relevant index of immune activation and the inflammatory response and more specifically of M1 macrophage and T helper (Th-1, Th-2, Th-17) activation. [73]. Although we used an adequate index of average blood sugar concentrations over the last few months (HbA1c) in conjunction with FBG, no independent effects could be established, although HbA1c was increased in AIS as compared with controls. Nevertheless, we accounted for the neurotoxic effects of glucose [19] by entering HbA1c in the neurotoxicity composite index computed here.

### 4.2. Antioxidants in AIS

In the current study, -SH groups, CMPAase activity and HDL, and a composite index of the CMPAase-HDL complex, were significantly lower in AIS patients as compared with controls and predicted baseline NIHSS and mRS scores. Previous research has demonstrated a negative relationship between total PON 1 activity and AIS [74,75,76,77]. Some studies found that serum PON1 activity is associated with the functional prognosis of AIS patients and that the mRS score of AIS patients tends to decline as serum PON1 activity levels increase [75,76]. Not all studies, however, found a significant association between PON1 activity and acute AIS [78]. However, we found that CMPAase activity rather than AREase activity was strongly and inversely associated with AIS and disabilities. Previously, Michalak et al. [79] showed that the ratio of paraoxonase/AREase was decreased in AIS and was associated with increased disability scores at baseline, 3, 6 and 12 months later. Lowered paraoxonase activities in association with lowered HDL and -SH groups and AIS were also established by Kotur-Stevuljevic et al. [80].

Individuals with low HDL are more likely to suffer a stroke or other cardiovascular incidents, indicating that HDL protects against AIS [81,82,83]. In this respect, it is important to note that PON1 activity, especially CMPAase activity, is strongly associated with HDL and in fact protects HDL and LDL from oxidation [51,52]. Interestingly, in our study, the CMPAase + HDL composite score was inversely associated with LOOH and with the neurotoxicity index, indicating that lowered protection against lipid peroxidation is indeed accompanied by increased lipid hydroperoxide levels. As such, the composite CMPAase + HDL, used in the present study, reflects the protective activities of the PON-HDL complex [51,52]. It is important to note that PON1 has three cysteine residues with functional features and that the thiol group at position 284 is a free thiol that partly defines PON’s antioxidant capabilities [84]. -SH groups can be replenished via reductive recycling by cell reductants such as the glutathione system [85]. Other papers have reported lowered levels of different antioxidant biomarkers in AIS including total antioxidant potential, cholesterol-adjusted carotenoids, vitamin E and vitamin D3, whilst most studies have not reported changes in -SH groups [19,20,62,86].

### 4.3. Oxidative and Antioxidant Markers Predict AIS Outcome

The current study established that a combination of lowered antioxidant defenses and increased LOOH production and neurotoxicity at baseline independently predicts the disabilities 3 and 6 months later and, in fact, more strongly predicts short-term outcome than the immune and metabolic biomarkers and classical risk factors as well. Moreover, we found a significant increase in NOx levels and a decrease in total PON1 activity and LOOH levels from baseline to three months later, whilst increases in AOPP from baseline to three months later were one of the important predictors of mRS scores 6 months after admission. Alexandrova et al. [15] showed elevated oxidative damage to lipids (increased LOOH) and spontaneous phagocyte oxidative activity (increased ROS production) in the chronic phase after stroke and found a strong increase in ROOH especially 4–9 months after AIS. Elsayed et al. [87] showed that baseline MDA levels assessed on admission correlated with clinical outcomes 3 months later. In addition, post-stroke death at 6–12 months was predicted by increased LOOH levels in conjunction with lowered antioxidant levels (25(OH)D), metabolic (FBG) and inflammatory (IL-6) biomarkers [20]. Biomarkers of oxidative stress, especially lipid peroxidation coupled with inflammatory biomarkers such as hsCRP, white blood cells, and vitamin D are biomarkers that not only predicted AIS but also prognosis and recurrence of AIS [19,20,65,68,88,89,90,91,92].

All in all, increased ROS/oxidative stress coupled with antioxidant defenses are associated with short-term outcomes of AIS and their impact is greater than that of inflammatory and metabolic biomarkers and classical risk factors as well.

### 4.4. Oxidative/Antioxidant Markers and Brain Imaging

In our study, increased neurotoxicity and lowered antioxidant defenses were also significantly associated with total DWI final infarct volume and FLAIR signal intensity, whist neurotoxicity, lowered antioxidant defenses, and both MRI measurements were independent predictors of a worse short-term outcome at 3 months. Our findings extend previous findings that an increased DWI lesion volume predicts a poor prognosis [93] and additionally show that adding neurotoxicity and antioxidant biomarkers further increases the prediction of mRS scores 3 months later. Previously, it was detected that in AIS, increased hyperacute plasma F2-isoprostane levels predict infarct volume and growth [94] and F2-isoprostanes and perchloric acid oxygen radical absorbance capacity (ORAC_PCA_) predict diffusion-perfusion mismatch and mismatch salvage [92]. F2-isoprostanes reflect the production of ROS-induced lipid peroxidation of membrane lipoproteins and phospholipids, whilst ORAC_PCA_ is a biomarker of total antioxidant defenses [94]. On the other hand, Kang et al. [95] did not detect any differences in PON1 activity between AIS patients with and without white matter hyperintensities. Recently, a Chinese research group [96] developed a new probe in AIS animal models to visualize oxidative stress levels (namely a highly ROS-responsive radiometric near-infrared-II nanoprobe) in the lesion through recognition of the impaired blood−brain barrier and activated endothelial cells. This highly ROS-based method allowed one to visualize oxidative stress levels in hyperglycemia mice with AIS and showed that hypergycemia mice have higher oxidative stress than normoglycemia mice after ischemic stroke, indicating increased peripheral LOOH and lowered antioxidant enzyme levels [96].

In animal models, it has been shown that free radical-induced oxidative stress and lowered antioxidant defenses are key contributors to the development of ischemic lesions, increasing infarct core volumes with more neurological disabilities, and reperfusion-related brain tissue damage [97,98,99]. It was hypothesized that the rapid increase in reactive oxygen species following AIS and the ensuing oxidative stress damage may contribute to neuronal cell and neurovascular unit injuries [13,100,101]. The key role of nitro-oxidative stress in the progression of the core lesions [97] is further corroborated by experiments based on blocking these pathways. Thus, in models of cerebral ischemia, catalase and superoxide dismutase (two antioxidant enzymes) and inhibition of neuronal nitric oxide synthase reduce infarct size [102,103].

Increased ROS formation, particularly in people with low antioxidant defenses, promotes irreversible neuronal loss due to necrotic cell death in the core, whereas salvageable penumbra cells may be further damaged by oxidative stress, thereby increasing apoptosis in the infarct core [11,19,20,104]. Activated oxidative pathways not only contribute to neuronal injury and neuronal death and tissue degradation but also contribute to BBB breakdown thereby allowing more M1 macrophages, activated T cells, effector cells, pro-inflammatory cytokines, and other neurotoxic products to flow through [11,15,20,105]. During ischemia, the breakdown of the BBB is accompanied by vasogenic cerebral edema, which affects white matter. On the other hand, the production of hydroxyl radicals and peroxynitrite contributes to cytotoxic or cellular edema [13].

### 4.5. PON1 Gene and AIS

Another major finding of this study is that—using PLS path analysis, which considered multilevel and multimediated pathways—the PON1 RR genotype has an overall significant protective effect on AIS and its disabilities. More specifically, the PON1 additive genetic model (coding RR as 2, QR as = 1 and QQ as 0) increases CMPAase activity and overall antioxidant defences, and decreases LOOH, neurotoxicity, brain DWI/FLAIR lesions volume and AIS outcome. The PON1 QQ genotype, on the other hand, significantly increases LOOH, neurotoxicity and brain lesion volume, thereby inducing a worse AIS outcome.

In a systematic review and meta-analysis of AIS and PON1 gene polymorphisms, Liu et al. [106] discovered that the R allele or RR genotype of the PON1 Q192R polymorphism was associated with an increased risk of AIS in the general population. In a Turkish community, the homozygous QQ genotype may be protective against AIS [107]. However, other studies show that the QQ or RR genotypes are risk factors for AIS [108,109]. More studies have not corroborated the findings on associations between PON1 genotypes and AIS [21,110,111,112]. Furthermore, the review by Liu et al. [106] was criticized for inconsistencies [113]. Both the RR and QQ genotypes may have beneficial effects for AIS; the Q isoform can inhibit LDL oxidation and hydrolyze lipid peroxides more effectively than the R isoform although the RR genotype can hydrolyze paraoxon faster [114,115,116,117,118].

These inconsistent findings may be explained by our results that the genotype–AIS relationship is multidimensional, with several potential mediated pathways leading to AIS, which sometimes have different signs, namely positive versus negative effects. As a result, simple association studies involving this PON1 gene (and many more genes in general) may yield false positive and negative results depending on the statistical models used, namely PLS with mediation versus simple association studies. Only when the distinct catalytic sites (CMPAase and AREase) and the molecules and functions that they influence (HDL, LOOH, neurotoxicity, brain lesion volumes) are included in multilevel mediation analysis can more genuine connections be determined.

All-in-all, our data imply that the PON1 gene has a major impact on multiple pathways that lead to AIS disabilities, and that the final association computed between the genotype and outcome is dependent on which pathways and target functions are examined in the model. It appears that an additive model, which implies a significant contribution of the QR genotype, is the best model to use in multilevel, multimediation PLS models.

### 4.6. Limitations

First, it would have been more interesting if we had also measured the amounts of neurotoxic M1 and Th-1 cytokines in this investigation. Second, it would have been more interesting if the biomarkers had been examined at multiple time periods after stroke to capture biphasic responses between 8 hours and two weeks following AIS. Third, while we segmented FLAIR signal intensity in MRI, this method does not allow us to differentiate adequately between WMH (as a consequence of small vessel disease or atherosclerosis) and (subcortical) stroke-related injuries. The current study’s strength is that we incorporated different biomarkers into regression analyses, resulting in a more accurate prediction of outcome variables. Furthermore, we identified the independent impacts of oxidative/antioxidant biomarkers above and beyond the effects of immunological, metabolic, and traditional risk factors. It should be underscored that the review of the PON1 literature is quite often hampered by insufficient information on the type of substrates used [74].

Furthermore, a prior protein–protein interaction network analysis based on the genetic variants determined in AIS and death due to AIS, indicated the underlying immune pathways, including Toll-Like Receptor 2/4, JAK-STAT, tumor necrosis factor-α, NOD-like, and other cytokine pathways [11]. As shown in another review, these pathways are strongly associated with reactive oxygen species and oxidative stress pathways [119]. This indicates that pathological interlinks between the products of antioxidant and immune genes and pathways and the ensuing disorders in these intertwined pathways predispose towards AIS.

## 5. Conclusions

Increased lipid peroxidation and lowered lipid-associated antioxidant defenses independently predict stroke outcome. The PON1 Q192R gene has multiple effects on stroke outcome which are mediated by its effects on lipid peroxidation, antioxidant defenses, neurotoxicity, DWI stroke volume and FLAIR signal intensity. Elevated lipid peroxidation and decreased activity of the PON1-HDL complex and -SH groups are therapeutic targets for the prevention of AIS and subsequent neurodegenerative processes as well as increased levels of oxidative reperfusion mediators owing to ischemia-reperfusion injury.

## Figures and Tables

**Figure 1 antioxidants-12-00188-f001:**
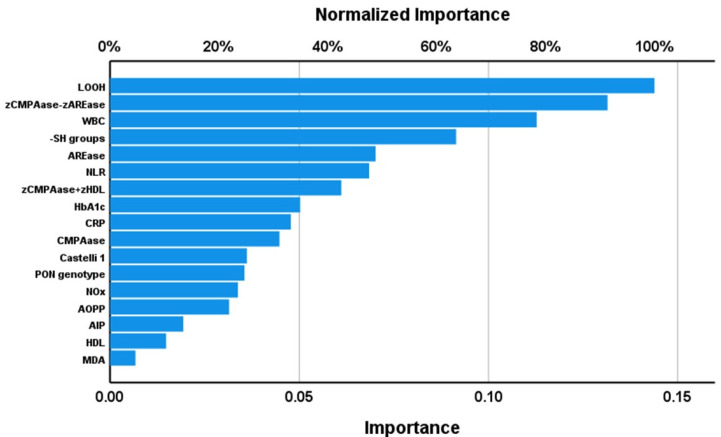
Importance chart of a neural network analysis with acute ischemic stroke (AIS) versus controls as input variables. LOOH: lipid hydroperoxides; zCMPAase: z transformation of chloromethyl acetate (CMPA)ase; zAREase: z transformation of aryl esterase; WBC: white blood cells; -SH: sulfhydryl; NLR: neutrophil/lymphocyte ratio; zHDL: z transformation of high-density lipoprotein cholesterol; HbA1c: glycated hemoglobin; CRP: C-reactive protein; Castelli 1: an index of the Castelli risk index 1; PON: paraoxonase; NOx: nitric oxide metabolites; AOPP: advanced oxidation protein products; AIP: and index of the atherogenic index of plasma; MDA: malondialdehyde.

**Figure 2 antioxidants-12-00188-f002:**
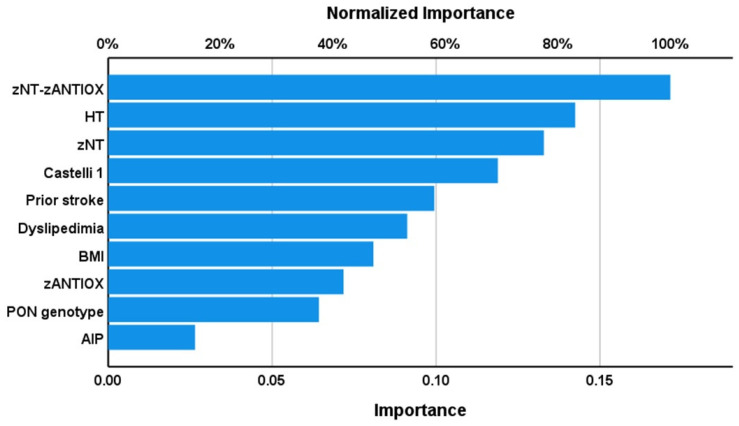
Importance chart of a neural network analysis with acute ischemic stroke (AIS) versus controls as input variables and composite scores of the predictors as input variables. zNT—z composite score of the neurotoxic analytes assayed in our study; HT—hypertension; zANTIOX—z composite score of the antioxidants assayed in our study; Castelli 1—an index of the Castelli risk index 1; BMI—body mass index; PON—paraoxonase; AIP—and index of the atherogenic index of plasma.

**Figure 3 antioxidants-12-00188-f003:**
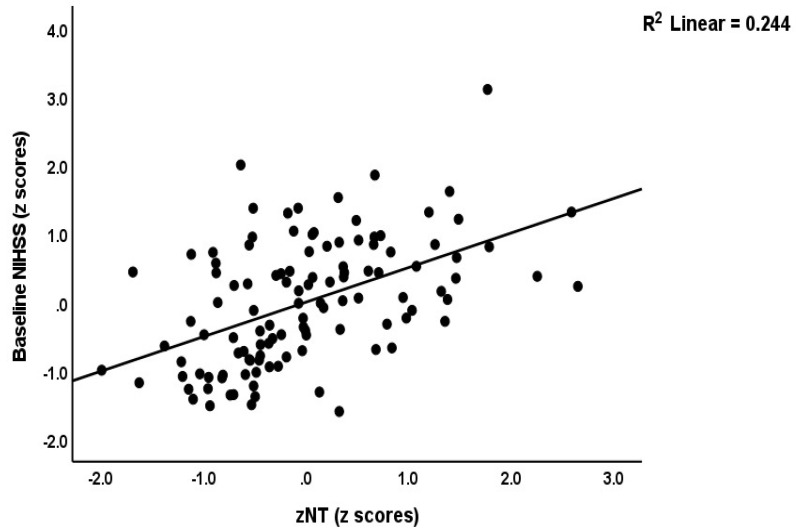
Partial regression of the baseline National Institutes of Health Stroke Scale (NIHSS) score on the neurotoxicity (zNT) index (after adjusting for age, sex, BMI, and smoking).

**Figure 4 antioxidants-12-00188-f004:**
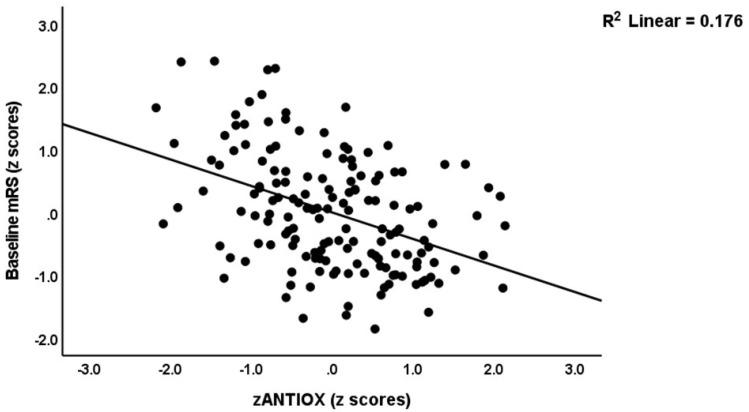
Partial regression of the baseline modified Rankin score (mRS) on an index of antioxidant defences (after adjusting for age, sex, BMI, and smoking).

**Figure 5 antioxidants-12-00188-f005:**
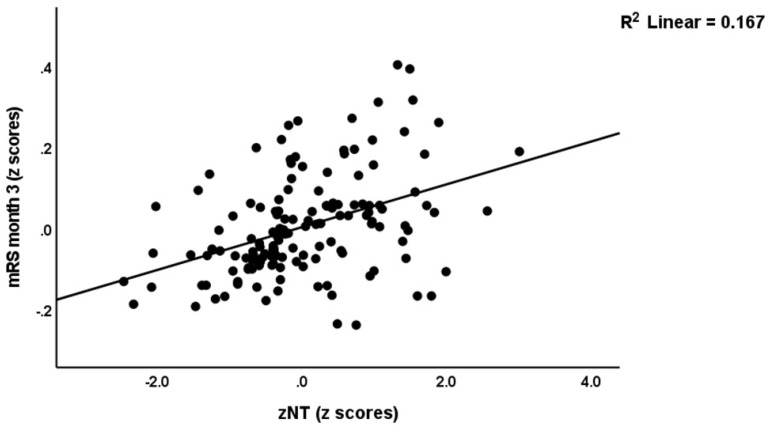
Partial regression of the modified Rankin score (mRS) at 3 months on the neurotoxicity index (zNT) after adjusting for age, sex, BMI, and smoking.

**Figure 6 antioxidants-12-00188-f006:**
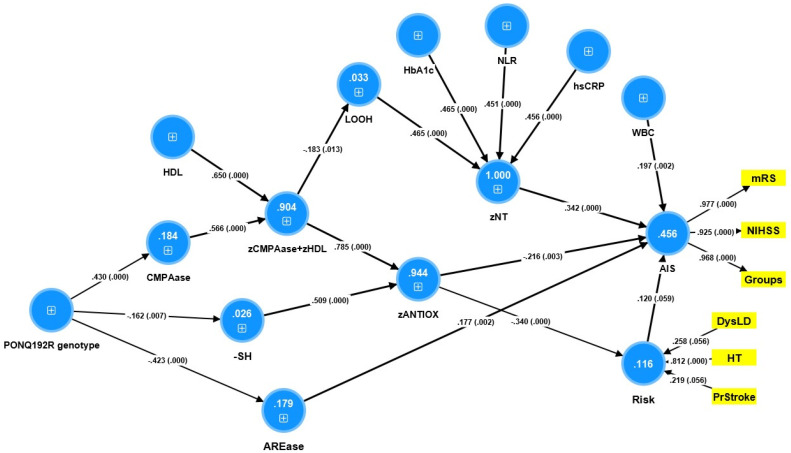
Results of partial least squares (PLS) analysis with severity of acute ischemic stroke (AIS) as final outcome variable. The direct explanatory variables are the neurotoxicity (zNT) index, white blood cell (WBC) count, antioxidant defenses (zANTIOX), classical risk factors (RISK), paraoxonase 1 (PON1) status, including chloromethyl phenylacetate CMPAase and arylesterase (ARE)ase activity and the PON1 Q192 genotype (entered as an additive model coding RR as 2, QR as 1 and QQ 0), high density lipoprotein cholesterol (HDL), and the CMPAase + HDL complex. zNT—composite based on C-reactive protein, neutrophil/lymphocyte ratio (NLR), lipid hydroperoxides (LOOH) and glycated hemoglobin (HbA1c). zANTIOX—a composte based on HDL, CMPAse and sulfhydryl (-SH) groups. AIS was conceptualized as a reflective model with the modified Rankin score (mRS) and National Institutes of Health Stroke Scale (NIHSS) scores as well as an ordinal variable based on the diagnostic groups shown in Table 1 as manifestations. DysLD—dyslipidemia, HT—hypertension, PrStroke—prior stroke.

**Table 1 antioxidants-12-00188-t001:** Clinical and biochemical data of healthy controls (HC) and patients with acute ischemic stroke (AIS) divided into those with mild (Mild AIS) and moderate (Moderate AIS) severity and healthy controls (HC).

Variables	HC ^A^ (*n* = 40)	Mild AIS ^B^ (*n* = 85)	Moderate AIS ^C^ (*n* = 37)	F/χ^2^	df	*p*
Age (years)	60.63 ± 9.18	59.84 ± 9.01	60.65 ± 9.46	0.16	2/159	0.857
Education (years)	4.58 ± 1.97	4.87 ± 1.98	4.16 ± 1.98	1.71	2/159	0.185
BMI (kg/m^2^)	24.90 ± 3.94	24.9114 ± 3.82	25.00 ± 3.89	0.01	2/159	0.994
Sex (Male/Female)	17/22	49/36	19/18	2.16	2	0.340
TUD (no/yes)	40/0	81/4	36/1	2.04	2	0.361
HT (no/yes)	-	26/59	14/23	0.62	1	0.433
T2DM (no/yes)	-	54/31	28/9	1.73	1	0.189
Dyslipidemia (no/yes)	-	56/29	26/11	0.23	1	0.635
Previous stroke (no/yes)	-	71/14	28/9	1.04	1	0.308
TOAST LAC/CEI/LAAS/U	-	11/4/46/10	11/3/18/2	FFHET	-	0.146
NIHSS + mRS index (z score)	−1.288 ± 0.00 ^B,C^	0.129 (0.395) ^A,C^	1.363 (0.594) ^A,B^	KWT	-	<0.001
NIHSS-basal	0.00 ± 0.00 ^B,C^	2.20 ± 1.00 ^A,C^	5.03 ± 1.94 ^A,B^	KWT	-	<0.001
mRS-basal	0.00 ± 0.00 ^B,C^	1.73 ± 0.50 ^A,C^	3.22 ± 0.42 ^A,B^	KWT	-	<0.001
mRS-3 months	0.00 ± 0.00 ^B,C^	0.89 ± 0.88 ^A,C^	1.65 ± 1.20 ^A,B^	KWT	-	<0.001
mRS-6 months	0.00 ± 0.00 ^B,C^	1.08 ± 1.41 ^A^	1.04 ± 1.55 ^A^	KWT	-	<0.001
hsCRP * (mg/L)	1.69 ± 1.05 ^C^	3.87 ± 9.63 ^C^	8.12 ± 15.35 ^A,B^	9.59	2/159	0.019
White blood cell count (K/µL)	5.34 (0.71) ^B,C^	7.91 (2.30) ^A,C^	9.06 (3.18) ^A,B^	27.93	2/159	<0.001
Neutrophile/Lymphocyte (NLR) *	1.652 (0.401) ^C^	2.246 (1.151) ^C^	3.762 (3.782) ^A,B^	11.66	2/159	<0.001
zIMMUNE (z score)	−0.744 (0.376) ^B,C^	−0.006 (0.906) ^A,C^	0.817 (1.049) ^A,B^	32.60	2/159	<0.001
Basal blood glucose (mg/dL)	107.6 (29.0) ^B^	132.1 (57.3) ^A^	124.7 (52.0)	3.21	2/159	0.043
HbA1C (%)	5.94 (0.64) ^B^	7.11 (2.16) ^A^	6.72 (2.08)	5.35	2/159	0.006
Total cholesterol * (mg/dL)	187.9 (23.6)	187.8 (45.9)	202.1 (78.4)	0.32	2/159	0.726
HDL-cholesterol (mg/dL)	59.31 (10.24) ^B,C^	45.70 (12.97) ^A^	44.00 (12.37) ^A^	20.40	2/159	<0.001
Triglycerides (mg/dL)	113.3 (44.3)	139.9 (89.9)	137.6 (106.2)	0.46	2/159	0.632
zTC-zHDL (z score)	−0.696 (0.517) ^B,C^	0.153 1.039 ^A^	0.401 (0.950) ^A^	16.23	2/159	<0.001
zTG-zHDL (z score)	−0.571 (0.800) ^B,C^	0.156 1.060 ^A^	0.259 (0.818) ^A^	9.75	2/159	<0.001
FLAIR signal intensitity (mm^3^) *	4651 (5223) ^C^	12,653 (12,406) ^C^	29,569 (23,415) ^A,B^	11.31	2/57	<0.001
FLAIR signal/brain volume *	2989 (2.713) ^B,C^	9988 (10,319) ^A,C^	25,410 (20,451) ^A,B^	13.83	2/56	<0.001
Total DWI stroke volume (mm^3^)	0.0 ^C^	1095 (2410) ^C^	17,997 (21,005) ^A,B^	KWT	-	<0.001
zFLAIR + zDWI (z score)	−0.969 (0.581) ^B,C^	0.043 (0.808) ^A,C^	1.088 (0.567) ^A,B^	30.47	2/57	<0.001

Values are presented as mean ± SD; F—results of analysis of variance; χ^2^—results of contingency analysis; KWT—results of the Kruskal–Wallis one-way analysis of variance; FFHET—Fisher–Freeman–Halton Exact Test. Patients are divided into those with higher National Institutes of Health Stroke Scale (NHSS) and modified Rankin score (mRS) values. ^A,B,C^: results of multiple comparisons among group means. BMI—body mass index; TUD—tobacco use disorder; HT—hypertension; T2DM—type 2 diabetes mellitus; TOAST LAC/CEI/LAAS/U—lacunar infarction/cardioembolic infarction / large artery atherosclerosis/stroke of other determined aetiology or unknown aetiology; hsCRP—C-reactive protein with high sensitivity assay; zIMMUNE—z unit-based composite score based on z NLR + z white blood cells + z hsCRP; HbA1C—hemoglobin A1C; HDL—high-density lipoprotein; zTC-zHDL—z unit-based composite score reflecting Castelli 1 risk index; zTG-zHDL—z unit-based composite score reflecting the atherogenic index of plasma; FLAIR—fluid-attenuated inversion recovery; DWI—diffusion-weighted imaging; zFLAIR + zDWI—z unit-based composite score that assesses total lesions as assessed with FLAIR and DWI. * Processed in log transformation.

**Table 2 antioxidants-12-00188-t002:** Results of general linear models (GLM) which show the oxidative stress and antioxidant biomarkers in patients with acute ischemic stroke (AIS) divided into those with mild (Mild AIS) and moderate (Moderate AIS) severity and healthy controls (HC).

Variable	HC ^A^ (*n* = 40)	Mild AIS ^B^ (*n* = 85)	Moderate AIS ^C^ (*n* = 37)	F/χ^2^	df	*p*
-SH groups basal (µmol/L)	296.7 (59.0) ^B,C^	254.7 (63.0) ^A^	247.7 (66.8) ^A^	7.58	2/159	<0.001
-SH groups 3 months (µmol/L)	296.7 (59.0) ^B,C^	246.6 (71.7) ^A^	262.2 (55.5) ^A^	6.94	2/113	0.001
PON1 Q192R QQ/QR/RR	6/23/11	8/37/38	5/18/12	FFHET	-	0.343
CMPAase basal (U/mL)	38.20 (1.73) ^B,C^	34.65 (1.17) ^A^	34.17 (1.65) ^A^	3.12	2/153	0.047
CMPAase 3 month (U/mL)	38.75 (1.73) ^B,C^	31.60 (1.48) ^A^	28.79 (2.10) ^A^	7.66	2/106	<0.001
AREase basal (U/mL)	254.00 (16.58) ^B,C^	333.82 (12.41) ^A^	335.80 (17.56) ^A^	9.52	2/153	<0.001
AREase 3 month (U/mL)	244.11 (90.25) ^B,C^	199.43 (67.54) ^A^	196.67 (95.54) ^A^	10.33	2/153	<0.001
zCMPAase-zAREase (z score)	0.852 (0.741) ^B,C^	−0.254 (0.946) ^A^	−0.337 (0.777) ^A^	26.05	2/169	<0.001
zCMPAase + zHDL (z score)	−0.635 (0.806) ^B,C^	−0.153 (0.964) ^A^	−0.334 (0.993) ^A^	12.74	2/159	<0.001
LOOH basal (RLU)	14,452 (3930) ^B,C^	27,381 (17,969) ^A^	26,788 (13,781) ^A^	22.19	2/159	<0.001
LOOH 3 months (RLU)	14,452 (3930) ^B,C^	21,203 (11,701) A	22,655 (10,144) A	7.74	2/112	<0.001
AOPP basal (µmol/L/eq. cloramin T)	262.3 (195.1)	213.7 (128.2)	196.4 (112.9)	2.22	2/159	0.112
AOPP 3 months (µmol/L/eq. cloramin T)	262.3 (195.1)	251.8 (175.0)	179.0 (105.9)	1.99	2/113	0.141
MDA basal (µM/mg protein)	1.526 (0.465)	1.433 (0.415)	1.505 (0.423)	0.76	2/1591	0.469
MDA 3 months (µM/mg protein)	1.526 (0.465)	1.691 (0.764)	1.403 (0.599)	1.82	2/112	0.166
NOx basal (µmol/L)	7.42 (5.37) ^B,C^	5.48 (3.02) ^A^	5.41 (2.23) ^A^	4.46	2/1591	0.013
NOx 3 months (µmol/L)	7.42 (5.37)	7.61 (4.32)	7.00 (4.57)	0.14	2/133	0.873
NT (z score)	−0.889 (0.442) ^B,C^	0.136 (1.002) ^A,C^	0.649 (0.749) ^A,B^	34.60	2/159	<0.001
ANTIOX (z score)	0.785 (0.736) ^B,C^	−0.195 (0.916) ^A^	−0.399 (0.994) ^A^	21.14	2/159	<0.001
NT/ANTIOX (z score)	−0.995 (0588) ^B,C^	−0197 (0.932) ^A,C^	0.623 (0.693) ^A,B^	43.89	2/159	<0.001

Values are presented as mean ± SD; F—results of analysis of variance; χ^2^—results of contingency analysis; KWT—results of the Kruskal–Wallis one-way analysis of variance; FFHET—Fisher–Freeman–Halton Exact Test. Patients are divided into those with higher National Institutes of Health Stroke Scale (NHSS) and modified Rankin score (mRS) values. ^A,B,C^: results of multiple comparisons among group means. -SH_ thiol groups; PON— Paraoxonase 1; CMPAase—chloromethyl phenylacetate ase activity; AREase—arylesterase activity; zCMPAase-zAREase—z unit-based composite score reflecting the difference between the two cat-alytic sites of the paraoxonase enzyme; HDL—high density lipoprotein; zCMPAase + zHDL—z unit-based composite score reflecting the protective CMPAase + HDL complex; LOOH—lipid hy-droperoxides; AOPP—advanced oxidation protein products; MDA—malondialdehyde; NOx—nitric oxide metabolites; zNT—z unit-based composite score reflecting neurotoxicity; zAN-TIOX—z unit-based composite score reflecting antioxidant capacities; zNT-zANTIOX—z unit-based composite score reflecting NT versus ANTIOX defenses.

**Table 3 antioxidants-12-00188-t003:** Results of two neural networks with acute ischemic stroke (AIS) versus healthy controls (HC) as output variables and oxidative stress and antioxidant biomarkers and clinical data as input variables.

	Models	NN#1 AIS versus HC	NN#2 AIS versus HC
Input Layer	Number of units	19	12
	Rescaling method	Normalized	Normalized
Hidden layers	Number of hidden layers	2	2
	Number of units in hidden layer 1	3	5
	Number of units in hidden layer 2	2	4
	Activation Function	Hyperbolic tangent	Hyperbolic tangent
Output layer	Dependent variables	AIS versus HC	AIS versus HC
	Number of units	2	2
	Activation function	Identity	Identity
	Error function	Sum of squares	Sum of squares
Training	Sum of squares error term	2.990	3.940
	% incorrect or relative error	5.5%	4.7%
	Prediction (sens, spec)	98.2%, 83.3%	93.4%, 100%
Testing	Sum of Squares error	0.843	0.634
	% incorrect or relative error	3.8%	3.7%
	Prediction (sens spec)	95.0%, 100%	100%, 66.7%
	AUC ROC	0.991	0.984
Holdout	% incorrect or relative error	8.5%	4.3%
	Prediction (sens, spec)	93.0%, 87.5%	97.0%, 92.3%

ROC—area under curve of receiver operating curve.

**Table 4 antioxidants-12-00188-t004:** Intercorrelation matrix.

Variable	NIHSS Basal	mRS Basal	mRS 3 Months	FLAIR Lesions	DWI Stroke Volume
-Sulfhydryl (SH) groups	−0.275 (<0.001)	−0.303 (<0.001)	−0.187 (0.028)	0.072 (0.589)	−0.290 (0.026)
CMPAase	−0.157 (0.046)	−0.203 (0.010)	−0.244 (0.003)	−0.443 (<0.001)	−0.483 (<0.001)
AREase	0.242 (0.002)	0.236 (0.003)	0.158 (0.060)	0.177 (0.188)	0.057 (0.666)
zCMPAase-zAREase	−0.379 (<0.001)	−0.417 (<0.001)	−0.373 (<0.001)	−0.548 (<0.001)	−0.492 (<0001)
HDL cholesterol	−0.321 (<0.001)	−0.418 (<0.001)	−0.366 (<0.001)	−0.320 (0.015)	−0.207 (0.112)
zCMPAase + zHDL	−0.271 (<0.001)	−0.353 (<0.001)	−0.377 (<0.001)	−0.489 (<0.001)	−0.393 (0.002)
Lipid hydroperoxides	0.292 (<0.001)	0.404 (<0.001)	0.294 (<0.001)	0.099 (0.466)	0.107 (0.416)
AOPP	−0.104 (0.192)	−0.140 (0.079)	0.033 (0.698)	−0.143 (0.288)	−0.304 (0.019)
Malondialdehyde	−0.059 (0.471)	−0.029 (0.721)	−0.025 (0.776)	−0.012 (0.929)	0.167 (0.211)
Nitric oxide metabolites	−0.149 (0.062)	0.041 (0.158)	−0.093 (0.274)	−0.056 (0.682)	−0.073 (0.581)
zNT	0.416 (<0.001)	0.538 (<0.001)	0.437 (<0.001)	0.347 (0.0085)	0.439 (<0.001)
zANTIOX	−0.362 (<0.001)	−0.477 (<0.001)	−0.393 (<0.001)	−0.506 (<0.001)	−0.474 (<0.001)
zNT-zANTIOX	0.462 (<0.001)	0.585 (<0.001)	0.481 (<0.001)	0.515 (<0.001)	0.554 (<0.001)
zIMMUNE	0.382 (<0.001)	0.529 (<0.001)	0.257 (0.002)	0.391 (0.003)	0.497 (<0.001)

NIHSS—National Institutes of Health Stroke Scale (NHSS); mRS—modified Rankin score (mRS). CMPAase—chloromethyl phenylacetate-ase activity; AREase—arylesterase activity; zCMPAase-zAREase—z unit-based composite score reflecting the difference between the two catalytic sites of the paraoxonase enzyme; HDL—high density lipoprotein; zCMPAase + zHDL—z unit-based composite score reflecting the protective CMPAase + HDL complex; AOPP—advanced oxidation protein products; zNT—z unit-based composite score reflecting neurotoxicity; zANTIOX—z unit-based composite score reflecting antioxidant capacities; zNT-zANTIOX—z unit-based composite score reflecting NT versus ANTIOX defenses; zIMMUNE—z unit based composite score reflecting immune activation; FLAIR lesion—lesions as assessed with fluid-attenuated inversion recovery (corrected for total brain volume); Total DWI stroke volume—infarction volume as measured by diffusion-weighted imaging.

**Table 5 antioxidants-12-00188-t005:** Results of multiple regression analysis with the National Institutes of Health Stroke (NIHSS) and modified Rankin score (mRS) scores as dependent variables and immune and oxida-tive biomarkers as explanatory variables.

Dependent Variable	Explanatory Variable	β	t	*p*	F_model_	df	*p*	R^2^
NIHSS basal	Model #1				11.36	6/143	<0.001	0.323
	HDL cholesterol	−0.164	−2.17	0.032		HDLc	−0.164	−2.17
	AREase	0.312	3.96	<0.001		AREase	0.312	3.96
	LOOH	0.182	2.55	0.012		LOOH	0.182	2.55
	-SH groups	−0.196	−2.78	0.006		-SH groups	−0.196	−2.78
	NLR	0.177	2.47	0.015		NLR	0.177	2.47
	CMPAase	−0.198	−2.44	0.016		CMPAase	−0.198	−2.44
mRS basal	Model #2				20.27	8/141	<0.001	0.535
	HT	0.128	1.87	0.064				
	HDLcholesterol	−0.152	−2.32	0.022				
	LOOH	0.248	4.05	<0.001				
	WBC	0.193	2.90	0.004				
	-SH groups	−0.183	−3.01	0.003				
	NLR	0.191	3.16	0.002				
	AREase	0.222	3.21	0.002				
	CMPAase	−0.180	−2.63	0.009				
NIHSS basal	Model #3				27.15	2/158	<0.001	0.256
	zNT-zANTIOX	0.367	4.90	<0.001				
	zCMPAaase-zAREase	−0.230	−3.06	0.003				
mRS basal	Model #4				43.00	45/153	<0.001	0.592
	zNT-zANTIOX	0.278	3.52	<0.001				
	HT	0.282	4.53	<0.001				
	zCMPAase-zAREase	−0.233	−3.77	<0.001				
	zIMMUNE	0.191	2.59	0.010				
mRS 3 months	Model #5				23.92	3/95	<0.001	0.430
	NT	0.475	5.87	<0.001				
	zCMPAase-zREase	−0.280	−3.47	<0.001				
	Previous stroke	0.166	2.13	0.036				
mRS 6 months	Model #6				7.03	4/85	<0.001	0.249
	Dyslipidemia	0.289	2.98	0.004				
	zCMPAase-zAREase	−0.227	−2.34	0.022				
	AOPP basal	−0.417	−3.014	0.003				
	AOPP 3 months	0.302	2.18	0.032				

HDL—high density lipoprotein; AREase—arylesterase activity; LOOH—lipid hydroperoxides; HDL c—HDL cholesterol; -SH—thiol groups; NLR—neutrophil/lymphocyte ratio; CMPAase—chloromethyl phenylacetate ase activity; HT—hypertensin; WBC—white blood cell count; zNT-zANTIOX—z unit-based composite score reflecting NT versus ANTIOX defenses; zCMPAase-zAREase—z unit-based composite score reflecting the equilibrium between two paraoxonase 1 (PON1) catalytic sites; zIMMUNE: zIMMUNE—z unit-based composite score reflecting immune activation based on z NLR + z white blood cell count + z hsC-reactive protein; PON RR—PON1; Q192R RR genotype; AOPP—advanced oxidation protein products.

**Table 6 antioxidants-12-00188-t006:** Results of multiple regression analysis with the National Institutes of Health Stroke (NIHSS) and modified Rankin score (mRS) scores as dependent variables, and immune and oxi-dative stress composite scores and brain imaging data as explanatory variables.

Dependent Variable	Explanatory Variable	β	t	*p*	F _model_	df	*p*	R^2^
NIHSS basal	Model #1				15.93	3/55	<0.001	0.465
	DWI Left Posterior	0.423	4.11	<0.001				
	zIMMUNE	0.323	3.15	0.003				
	zCMPAase-zAREase	−0.228	−2.16	0.035				
mRS basal	Model #2				24.32	4/53	<0.001	0.647
	zNT-zANTIOX	0.458	4.80	<0.001				
	HT	0.250	2.69	0.010				
	DWI Left Posterior	0.295	3.42	0.001				
	DWI Right Anterior	0.199	2.33	0.024				
mRS 3 months	Model #3				12.07	4/46	<0.001	0.512
	zNT-zANTIOX	0.304	2.54	0.015				
	DWI Right Anterior	0.281	2.65	0.011				
	Dyslipidemia	0.304	2.92	0.005				
	FLAIR signal intensity	0.284	2.36	0.022				
Total DWI	Model #4				16.71	2/55	<0.001	0.378
	zNT-zANTIOX	0.433	3.44	0.001				
	zCMPAase-zAREase	−0.263	−2.09	0.042				
FLAIR signal intensity	Model #5				15.61	3/52	<0.001	0.474
	zCMPAase-zAREase	−0.343	−2.86	0.006				
	Previous stroke	0.322	3.12	0.003				
	zNT-zANTIOX	0.272	2.26	0.028				

DWI—diffusion-weighted imaging; zIMMUNE—z unit-based composite score reflecting immune activation based on z NLR + z white blood cell count + z hsC-reactive protein; zNT-zANTIOX—z unit-based composite score reflecting neurotoxicity versus ANTIOX defenses; FLAIR—fluid-attenuated inversion recovery (corrected for total brain volume); PON1 RR—paraoxonase Q192R RR genotype.

## Data Availability

The dataset generated during and/or analyzed during the current study will be available from MM upon reasonable request and once the authors have fully exploited the dataset.

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
