# Peer review of "In Mild and Moderate Acute Ischemic Stroke, Increased Lipid Peroxidation and Lowered Antioxidant Defenses Are Strongly Associated with Disabilities and Final Stroke Core Volume"

_antioxidants, 2023, doi:10.3390/antiox12010188_

Round 1

Reviewer 1 Report

The original article titled: In mild and moderate acute ischemic stroke, increased lipid peroxidation and lowered antioxidant defenses are strongly associated with disabilities and final stroke core volume is written well with precise analysis. The biggest strength of this study is analysis of different biomarkers. However, the paper is too long you should focus on your results the discussion is like review. 

Author Response

x

REFEREE1

The original article titled: In mild and moderate acute ischemic stroke, increased lipid peroxidation and lowered antioxidant defenses are strongly associated with disabilities and final stroke core volume is written well with precise analysis. The biggest strength of this study is analysis of different biomarkers. However, the paper is too long you should focus on your results the discussion is like review. 

@ANSWER: My problem is that reviewer two requested many more discussion and reviews of the previous literature. Given this, I did not reduce the discussion, but instead added more reviews and references . In addition, I re-read my discussion as it stands and I think it is quite well focused, even with the additional reviews requested by referee 2.

Reviewer 2 Report

This paper is intented  to explore the possible link between the stroke severity and progression with the antioxidant defence of the affected brain tissue. The novelty of the study can be only partially recognized, several similar studies on animals as well as in human clinical settings can be found. Paper only marginally utilizes very recent literature data on this topic. Additional notes should be clarified. 1. Measurement of LOOH, MDA,  NOx, -SH groups is rather unspecific and can not give a relevant data on the affected tissue if detecetd in plasma. 2. Interpretation of the experimental results and relevant discussion lacks a deeper elaboration and pathological interlinks based on the latest studies. Paper is presented in a rather complex and also complicated form to clearly manifest a novelty and pertinence of the presented results.

in a deeper manner and pathological interlinks shoold be more elaborated based on the results of latest studies

Author Response

REFEREE 2

This paper is intented  to explore the possible link between the stroke severity and progression with the antioxidant defence of the affected brain tissue. The novelty of the study can be only partially recognized, several similar studies on animals as well as in human clinical settings can be found.

@ANSWER There are no studies that define the association between the PON1 genotypes, their products (enzyme activities) and oxidative stress pathways, and the affected tissues as assessed with DWI and FLAIR. This study is therefore unique.

Paper only marginally utilizes very recent literature data on this topic.

@ANSWER: up to 33% of the citations are papers published the last 5 years. So, that is is not “marginally” at all.

Additional notes should be clarified. 1. Measurement of LOOH, MDA,  NOx, -SH groups is rather unspecific and can not give a relevant data on the affected tissue if detecetd in plasma.

@ANSWER: Incorrect remark, because we showed that peripheral markers are associated with the damage to central tissues. These are the findings, please read  !!!!

  1. Interpretation of the experimental results and relevant discussion lacks a deeper elaboration and pathological interlinks based on the latest studies.

@ANSWER: According to referee 1 we even show too many links and reviews. Nevertheless, we added more review in the different sections:

Previously, Michalak et al. [79] showed that the ratio of paraoxomase / AREase was decreased in AIS and was associated with increased disability scores at baseline, 3, 6 and 12 months later. Lowered paraoxonase activities in association with lowered HDL and -SH groups and AIS were also established by Kotur-Stevuljevic et al.[80].

On the other hand, Kang et al. [95] did not detect any differences in PON activity between AIS patients with and without white matter hyperintensities.

It was hypothesized that the rapid increase in reactive oxygen species and ensuing oxidative stress following AIS may contribute to neuronal cell and neurovascular unit injuries [100-101].

However, other studies show that the QQ or RR genotypes are risk factors for AIS [108,109]. More studies did not corroborate the findings on associations between PON1 genotypes and AIS [21,110,111;112]. Furthermore, the review by Liu et al. [106] was criticized for inconsistencies [113].

It should be underscored that the review of the PON1 literature is quite often hampered by insufficient information on the type of substrates used [119]. Furthermore, a prior protein-protein interaction network analysis based on the genetic variants determined in AIS and death due to AIS, indicated the underlying immune pathways, including Toll-Like Receptor 2/4, JAK-STAT, tumor necrosis factor-α, NOD-like, and other cytokine pathways [11]. A shown in another review, these pathways are strongly associated with reactive oxygen species and oxidative stress pathways [120]. This indicates that pathological interlinks between the products of antioxidant and immune pathways and the ensuing intertwined disorders in these pathways predispose towards AIS.

Paper is presented in a rather complex and also complicated form to clearly manifest a novelty and pertinence of the presented results.

@ANSWER: As was/is discussed in the text, such data should be analysed using PLS analysis with multi-step mediation. As discussed, not doing so leads to erroneous results. Is this complex?? Yes, for readers who are not familiar with the methods. Therefore, scientists in molecular neurobiology should finally start to comprehend and use these mediation methods.

in a deeper manner and pathological interlinks shoold be more elaborated based on the results of latest studies

@ANSWER: see above.

Round 2

Reviewer 2 Report

Authors thackled to explain most of my questions or nota, however, still there are several notesand questions which should be clarified. 1. For clinical generalization, explicit note for the patients origin and their genetic backgroud (different from indo-european, or others) should be added.

2. Specifically for this journal, it is reqiured that data expression are in a correct manner, see LOOH (per mg?), AOPP per mg?, NOx /l or mg ? many clinical studies explored a change in a total proteinemia in serum(plasma) after stroke

3.Discussion still misses information about the respected biological meanings  and representation of given parameters of N-O stress,  specifically in serum (which lipoproteins, protein? ,  their relevance to the pathomechanism of stroke). This is the same for the affected tissue, and what are the proposed functional consequenses on molecular pathophysiology of affected tissue?

According to that please restructuarize Abstract and Conclusions.

Author Response

Authors thackled to explain most of my questions or nota, however, still there are several notesand questions which should be clarified.

  1. For clinical generalization, explicit note for the patients origin and their genetic backgroud (different from indo-european, or others) should be added.

@ANSWER: added in the Methods section: Thai patients and controls

  1. Specifically for this journal, it is reqiured that data expression are in a correct manner, see LOOH (per mg?), AOPP per mg?, NOx /l or mg ? many clinical studies explored a change in a total proteinemia in serum(plasma) after stroke

@ANSWER: all data are expressed in the correct units (of course), as we and others have published repeatedly in other illnesses (MDD, BD, MS, Parkinson, epilepsy, schizophrenia ….).

3.Discussion still misses information about the respected biological meanings  and representation of given parameters of N-O stress,  specifically in serum (which lipoproteins, protein? ,  their relevance to the pathomechanism of stroke). This is the same for the affected tissue, and what are the proposed functional consequenses on molecular pathophysiology of affected tissue? According to that please restructuarize Abstract and Conclusions.

@ANSWER: In the Introduction we already explained: b) increased ROS and nitro-oxidative stress with elevated lipid peroxidation (lipid hydroperoxides or LOOH) and nitric oxide metabolite (NOx) levels,

In the Methods section we have now added:

AOPP measures oxidation-modified albumin, lipoproteins and fibrinogen). NO metabolites (NOx, namely nitrite and nitrate) were determined …

The molecular effects of ROS and nitro-oxidative stress and how they contribute to stroke were/are discussed. Referee 1 denoted this already as too long. It was and is stated in the discussion that:

In animal models, it has been shown that free radical–induced oxidative stress and lowered antioxidant defenses are key contributors to the development of ischemic lesions, increasing infarct core volumes with more neurological disabilities, and reperfusion–related brain tissue damage [97-99]. It was hypothesized that the rapid increase in reactive oxygen species following AIS and the ensuing oxidative stress damage may contribute to neuronal cell and neurovascular unit injuries [100-101]. The key role of nitro-oxidative stress in the progression of the core lesions [97] is further corroborated by experiments based on blocking these pathways. Thus, in models of cerebral ischemia, catalase and superoxide dismutase (two antioxidant enzymes) and inhibition of neuronal nitric oxide synthase reduce infarct size [102,103].

Increased ROS formation, particularly in people with low antioxidant defenses, promotes irreversible neuronal loss due to necrotic cell death in the core, whereas salvageable penumbra cells may be further damaged by oxidative stress, thereby increasing apoptosis in the infarctcore [11,19,20,104]. Activated oxidative pathways not only contribute to neuronal injury and neuronal death and tissue degradation but also contribute to BBB breakdown thereby allowing more M1 macrophages, activated T cells, effector cells, pro-inflammatory cytokines, and other neurotoxic products to flow through [11,15,20,105]. During ischemia, the breakdown of the BBB is accompanied by vasogenic cerebral edema, which affects the white matter. On the other hand, the production of hydroxyl radicals and peroxynitrite contributes to cytotoxic or cellular edema [13].

Furthermore, these effects were already stated at the end of the Abstract, as:

Lipid peroxidation and lowered -SH and PON1-HDL activity are drug targets to prevent AIS and consequent neurodegenerative processes and increased oxidative reperfusion mediators due to ischemia-reperfusion injury.